# The role of quantitative T1 and T2 mapping for detecting minimal hepatic encephalopathy in chronic hepatic schistosomiasis patients

**Xue-Fei Liu[☉], Ke-Ying Wang[☉], Hai-Feng Shi[☉], Ying Li[iD]\*, Xin Li\***

Department of Radiology, Jinshan Hospital of Fudan University, Shanghai, China

[☉] These authors contributed equally to this work
\* dr.yingli@foxmail.com (YL); 362990170@qq.com (XL)

## Abstract

### Background

Minimal hepatic encephalopathy (MHE) is a frequent neurocognitive complication in chronic hepatic schistosomiasis (CHS) patients. Conventional diagnostic tools are time-consuming and influenced by education level. Quantitative MRI mapping offers a potential non-invasive biomarker for MHE, but its role in CHS remains unclear.

### Aim

This study aimed to evaluate the diagnostic value of quantitative T1 and T2 mapping for detecting MHE in CHS patients.

### Methods

This prospective cross-sectional observational study was conducted from August 2023 to July 2024. A total of 88 CHS patients were enrolled and divided into MHE (n = 50) and non-MHE group (n = 38). MHE was assessed using the number connection test-A (NCT-A) and digit symbol test (DST). All participants underwent 3.0T MRI with T1 and T2 mapping. Group comparisons, correlation analyses, logistic regression, and receiver operating characteristic (ROC) curve analyses were performed.

### Results

MHE patients exhibited significantly lower T1 values across multiple brain regions, particularly the globus pallidus (all p < 0.001). T2 values showed no significant differences between groups. Multivariable regression confirmed that lower T1 values were independently associated with MHE (p = 0.009). ROC analysis demonstrated excellent diagnostic performance of globus pallidus T1 values (AUC = 0.92, 95% CI: 0.86–0.99), with sensitivity of 0.97 and specificity of 0.90. T1 values were also correlated with neuropsychological tests (NCT-A: r = −0.345, p < 0.001; DST: r = 0.232,

**Data availability statement:** The raw data supporting the conclusions of this article can be found at https://github.com/jsyyky/MHE.

**Funding:** This work was supported by Shanghai Jinshan District Science and Technology Committee (2023-WS-07) to Xin Li and by Jinshan Hospital (KYQDJJ202501) and Jinshan District Health Commission (JSZK2023A02) to Hai-Feng Shi.

**Competing interests:** The authors have declared that no competing interests exist.

$p = 0.029$). However, T2 values showed no significant group differences or diagnostic value.

## Conclusion

Quantitative T1 mapping, rather than T2 mapping, could be used as a potential non-invasive biomarker for detecting MHE in CHS patients. This approach offers an objective complement to psychometric testing and may facilitate earlier diagnosis and monitoring of MHE progression.

## Introduction

Chronic hepatic schistosomiasis (CHS) is a cause of hepatic encephalopathy (HE) second to cirrhosis [1]. There were about 11.6 million people infected by schistosomiasis japonicum and 100 million people at risk in China [2]. Patients with HE experience widely variable cognitive impairments that range from subtle neuropsychological impairment (minimal hepatic encephalopathy, MHE) to obvious neurophysiological abnormalities (overt HE) [3]. MHE can affect up to 80% of chronic liver disease patients due to the decreased detoxification function of the liver or the formation of portosystemic shunts [3]. In developing countries, particularly in schistosomiasis-endemic regions of Africa, Asia, and South America, MHE prevalence in CHS patients remains poorly characterized, with limited epidemiological data available. In China, studies suggest MHE affects approximately 40%−80% of CHS patients [4,5].

MHE significantly impairs quality of life, increases the risk of progression to overt HE, elevates mortality rates, and imposes substantial healthcare costs [3]. Early detection of MHE is therefore crucial for timely intervention and improved patient outcomes. Currently, psychometric hepatic encephalopathy score (PHES) is considered as the "gold" standard diagnosis for MHE, which includes 5 tests named number connection tests (NCT-A and NCT-B), line tracing test (LTT), digit symbol test(DST), and serial dotting test (SDT) [6]. However, these tests are quite time-consuming and the outcomes are influenced by the educational background [7].

Paraclinical examinations, including ammonemia measurement and brain imaging techniques, can contribute to the diagnostic process of MHE [8]. MRI T1-weighted hyperintensities in the basal ganglia are common in cirrhosis-related MHE patients, due to brain manganese deposition caused by portosystemic shunting and bypassing liver metabolism of manganese [9]. Basal ganglia T1-weighted hyperintensities were also reported in CHS-related MHE patients, who usually develop into type B MHE and is associated with portal hypertension and portosystemic shunting in the absence of intrinsic liver dysfunction and hyperammonemia [4,9]. Conventional T1-weighted imaging only provides a qualitative method based on signal intensity, relying on radiologists' subjective interpretation. Thus, T1-weighted hyperintensities are not part of routine diagnostic criteria for MHE. The development of objective, non-invasive biomarkers for MHE detection represents a critical need.

Quantitative T1 mapping offers objective, reproducible assessment of T1 relaxation times. Thus, T1 mapping has become a valuable tool for detecting brain changes in patients with HE [10]. T1 mapping was reported that could detect changes in brain regions, especially the globus pallidus, correlating with the severity of HE. Lower T1 values were reported in HE rats, which was correlated with brain manganese content [11]. T1 mapping has shown promise in cirrhosis and animal models. However, the application of T1mapping in MHE of CHS patients has not been reported yet.

We assumed that that quantitative T1 mapping, especially in the globus pallidus, may discriminate between CHS patients with and without MHE. Furthermore, quantitative T2 mapping was exploratorily used to detect subtle tissue changes to see if T2 changes could aid MHE detection. Thus, the aim of this study is to evaluate whether a MRI T1 and T2 mapping could be used for clinical diagnoses in MHE in CHS patients.

## Materials and methods

### Ethical considerations

The research adhered to the ethical principles set forth in the Declaration of Helsinki. Ethical clearance was granted by the Institutional Review Board of Jinshan Hospital (JIEC2023-S66). Informed written consent was acquired from every participant prior to their inclusion in the study. The participants were provided with comprehensive information regarding the study's aims, methodologies, and the potential risks and benefits involved of MRI examination. To safeguard participant confidentiality, patient data was anonymized, and the database was secured with restricted access measures.

### Study design and determination of sample size

This research was structured as a prospective cross-sectional observational study aimed at investigating the relationship between quantitative T1 and T2 mapping and the likelihood of developing MHE in CHS patients. The calculation of the sample size was conducted via a post – hoc power analysis of the results of this study. It assumed a moderate effect size (Cohen's d = 0.7) to identify a statistically significant difference in T1 measurements between patients with or without MHE. By employing a two-tailed test with a significance threshold (α) set at 0.05 and a statistical power (1-β) of 0.80, it was determined that each group would require approximately 36 participants.

### Data acquisition

From August 15, 2023 and July 15, 2024, consecutive CHS patients referred to the local hepatology clinics and the gastroenterology department were screened for eligibility. All patients were invited to participate if they had a confirmed diagnosis of CHS and were willing and able to undergo MRI examination and psychometric testing. The MHE status was determined after enrollment based on psychometric test results, allowing classification into MHE and non-MHE groups for comparative analysis. The exclusion criteria were: (1) history of cirrhosis, alcoholism, and hepatocellular carcinoma; (2) using any drugs with liver or central nervous system toxicity; (3) contraindications to MRI; (4) obvious artifacts on MRI. Medical history, age, and education level were collected using a structured questionnaire at the time of enrollment.

### Diagnosis of chronic hepatic schistosomiasis

The determination of chronic hepatic schistosomiasis relied on an integration of clinical, laboratory, and imaging assessment methods. A patient's clinical history was scrutinized for indications of exposure to areas where Schistosoma is prevalent or for a history of prior schistosomiasis treatment. Laboratory verification was achieved through serological tests aimed at identifying antibodies targeting Schistosoma species. Imaging examinations, including abdominal ultrasound or CT scans, were employed to evaluate features indicative of CHS, such as periportal fibrosis, liver calcifications, splenomegaly, or the development of collateral vessels.

## Assessment of minimal hepatic encephalopathy

The identification of MHE was grounded in the outcomes of the number connection test-A (NCT-A) and the digit symbol test (DST). A diagnosis of MHE was established when both of the two tests showed abnormal results. According to previously reported criteria, the combined use of NCT-A and DST demonstrated a sensitivity of 76.9% and a specificity of 96.3% in diagnosing MHE within the Chinese population [12].

In the NCT-A test, patients were presented with a sheet containing 25 circles, each numbered from 1 to 25, randomly distributed across the page. The task required patients to connect the circles in ascending numerical order as quickly and accurately as possible. The examiner recorded the total time in seconds required to complete the task. Errors must be corrected immediately before continuing. NCT-A scores ≥ 50 seconds were considered abnormal and indicative of impaired psychomotor speed and executive dysfunction.

In the DST test, patients were provided with a key showing nine unique symbols paired with digits 1–9 at the top of the test sheet. Below this key, rows of randomly ordered digits were presented with empty boxes beneath each digit. The task required patients to fill in the corresponding symbol for each digit as quickly and accurately as possible within a 90-second time limit. The score represented the total number of correctly completed symbol-digit pairs. DST scores ≤ 39 correct responses were considered abnormal.

Psychometric tests were administered by trained examiners (XFL and KYW). Participants undertook the tests autonomously, after receiving clear and consistent verbal instructions from the examiner. No aid was offered during test completion to preserve the authenticity of the participant's cognitive assessment. The complete psychometric evaluation, including instructions, required approximately 10–15 minutes per patient. All psychometric tests were completed on the same day as MRI scanning.

## Laboratory tests

Laboratory tests were carried out on the same day as the MRI scanning. All these laboratory assessments were aimed at evaluating liver function, including alanine aminotransferase (ALT), aspartate aminotransferase (AST), total bilirubin (TB), albumin (ALB), prothrombin time (PT), international normalized ratio (INR), and platelet count (PLT).

## Magnetic resonance imaging protocol

MRI examinations were conducted using a 3.0 Tesla Siemens system (uMRI 780, United Imaging, Shanghai) equipped with advanced quantitative mapping capabilities to assess the comprehensive characterization of brain regions.

T1 mapping sequence was used for quantitative assessment of longitudinal relaxation times, enabling measurement of native T1 values, which reflects the tissue's ability to recover magnetization after radiofrequency excitation. T1 mapping was particularly valuable for detecting metal (manganese) deposition that correlates with portosystemic shunting severity and neurocognitive impairment. Imaging parameters included repetition times (TR) of 200 or 800 ms with corresponding inversion times (TI) optimized for T1 recovery, an echo time (TE) of 1.2 ms, and a flip angle of 35°. The field of view (FOV) was set to 340 × 340 mm with a matrix size of 256 × 256, slice thickness of 8 mm.

T2 mapping sequence was used for quantitative evaluation of transverse relaxation times, enabling measurement of native T2 values, which was sensitive to tissue water content and microstructural changes. T2 mapping could detect increased free water content from low-grade cerebral edema due to astrocyte dysfunction. A multi-echo spin-echo sequence was employed with eight TEs ranging from 10–80 ms (ΔTE = 10 ms), TR of 1000 ms, and a flip angle of 90°. Identical geometric parameters were used with FOV of 340 × 340 mm, matrix size of 256 × 256, and slice thickness of 8 mm.

Quantitative T1 and T2 values were obtained by manually delineating regions of interest (ROIs) in the frontal, temporal, and occipital lobes, caudate nucleus, and globus pallidus. Two experienced radiologists (XFL and XL), blinded to clinical, independently measured T1 and T2 values, with discrepancies resolved by consensus. The mean of the two measurements was used for statistical analysis.

The area under the receiver operator characteristic (ROC) curve (AUC) with 95% confidence interval (CI) were used to evaluate the discrimination of the MRI parameters in diagnosis of MHE. The sensitivity, specificity, positive predictive value (PPV), and negative predictive value (NPV) were calculated.

## Statistical analysis

All statistical analyses were performed utilizing R software (version 4.5.0). Continuous variables were presented in the format of mean ± standard deviation. The normality of all continuous variables was assessed using the Shapiro-Wilk test. Based on the data distribution, group comparisons for continuous variables were conducted using either Student's t-test (for normal distribution data) or the Mann-Whitney U test (for non-normally distributed data). For categorical variables, comparisons were made using Chi-square tests. Univariate analyses were first performed to identify variables potentially associated with MHE. Variables with $p < 0.05$ in univariate analyses, along with clinically relevant covariates (age, gender, and education level), were entered into a multivariable logistic regression model to determine independent predictors of MHE. To explore the associations between T1 and T2 values and MHE, Pearson or Spearman correlation analyses were carried out according to the data distribution. ROC curve analysis was applied and the AUC was used as an indicator of diagnostic performance of MHE. A p-value of less than 0.05 was considered statistically significant throughout the analyses.

## Results

### Demographic and clinical characteristics

This study encompassed 88 patients diagnosed with CHS, who were categorized into two groups: 50 patients with MHE and 38 patients without MHE (non-MHE). The gender distribution was comparable between the two groups, with females accounting for 38.0% in the MHE group and 44.7% in the non-MHE group ($p = 0.676$). The mean age was also similar across the groups, with the MHE group having a mean age of 70 ± 6.5 years (range: 50–84 years) and the non-MHE group having a mean age of 69 ± 7.9 years (range: 47–84 years) ($p = 0.365$). The clinical and laboratory characteristics, as well as T1 and T2 values, for both MHE and non-MHE cases are presented in Table 1 and S1 Table. The workflow of this study is illustrated in Fig 1.

### NCT-A and DST tests

The NCT-A scores were significantly higher in the MHE group compared to the non-MHE group ($p < 0.001$). Conversely, the DST scores were lower in the MHE group when compared to the non-MHE group ($p < 0.001$).

### Liver function and biochemical markers

Patients with MHE had significantly elevated AST ($p = 0.043$) and TB ($p = 0.054$) levels compared to those in the non-MHE group. However, there were no significant differences in ALT ($p = 0.550$), PT ($p = 0.358$), INR ($p = 0.960$), and PLT ($p = 0.212$) levels between the two groups.

### T1 and T2 values

Significant differences in T1 values were observed between MHE and non-MHE groups across multiple brain regions. MHE patients demonstrated significantly lower T1 values compared to non-MHE patients in the frontal lobe ($p = 0.001$), temporal lobe ($p = 0.003$), occipital lobe ($p = 0.011$), caudate nucleus ($p < 0.001$), and globus pallidus ($p < 0.001$).

In contrast, T2 values revealed no significant differences between groups, including the frontal lobe ($p = 0.938$), temporal lobe ($p = 0.180$), occipital lobe ($p = 0.226$), caudate nucleus ($p = 0.682$), and globus pallidus ($p = 0.149$) between MHE and non-MHE groups. A case of T1 and T2 mapping is shown in Fig 2.

 

**Table 1. The clinical and laboratory characteristics and T1 and T2 values in MHE and non-MHE cases.**

| Parameters | MHE (N = 50) | non-MHE(N = 38) | P-value |
|---|---|---|---|
| Gender | | | 0.676 |
| Female | 19 (38.0%) | 17 (44.7%) | |
| Male | 31 (62.0%) | 21 (55.3%) | |
| Education level | | | |
| Above primary school | 5 (10.0%) | 4 (10.5%) | 1.000 |
| Below primary school | 45 (90.0%) | 34 (89.5%) | |
| Age (y) | 70.7 ± 6.59 | 69.3 ± 7.99 | 0.365 |
| NCT-A | 57.7 ± 16.4 | 37.7 ± 9.38 | <0.001 |
| DST | 34.3 ± 13.8 | 45.4 ± 13.5 | <0.001 |
| AST (U/L) | 46.9 ± 22.6 | 37.7 ± 19.4 | 0.043 |
| T1 Frontal Lobe | 1130 ± 192 | 1300 ± 257 | 0.001 |
| T1 Temporal Lobe | 1690 ± 393 | 1910 ± 302 | 0.003 |
| T1 Occipital Lobe | 1240 ± 305 | 1400 ± 271 | 0.011 |
| T1 Caudate Nucleus | 1390 ± 246 | 1890 ± 349 | <0.001 |
| T1 Globus Pallidus | 1250 ± 324 | 1790 ± 315 | <0.001 |
| T2 Frontal Lobe | 96.9 ± 61.0 | 95.8 ± 70.1 | 0.938 |
| T2 Temporal Lobe | 102 ± 24.7 | 95.8 ± 18.6 | 0.180 |
| T2 Occipital Lobe | 93.9 ± 11.7 | 91.6 ± 5.06 | 0.226 |
| T2 Caudate Nucleus | 79.9 ± 9.39 | 79.1 ± 8.19 | 0.682 |
| T2 Globus Pallidus | 72.0 ± 14.0 | 76.4 ± 14.3 | 0.149 |

ALB, albumin; ALT, alanine aminotransferase; AST, aspartate aminotransferase; DST, digit symbol test; INR, international normalized ratio; MHE, minimal hepatic encephalopathy; NCT-A, number connection test-A; PLT, platelet count; PT, prothrombin time; TB, total bilirubin

Continuous variables are presented as the mean ± standard deviation, and categorical variables are presented as frequency values in numbers and percentages (%).

## Multivariable adjustment and correlation analyses

Multivariable logistic regression analysis revealed that none of the potential confounders (age, gender, education level, AST, T1 frontal lobe, T1 temporal lobe, T1 occipital lobe, and T1 caudate nucleus values) were independently associated with MHE after adjustment (all p > 0.05, Table 2). Lower T1 value in globus pallidus was strongly associated with an increased likelihood of MHE (p = 0.004).

Correlation analyses were performed to explore the relationships between MHE and T1 values. The Shapiro-Wilk test indicated that T1 and T2 values, NCT-A and DST scores, and ALB were normally distributed (all p > 0.05). Therefore, Pearson correlation was applied. However, AST, ALT, TB, PT, INR and platelet were non-normally distributed (p < 0.05). Therefore, Spearman correlation was used. There was a strong negative correlation between MHE status and T1 globus pallidus values (r = −0.644, 95% CI: −0.752 to −0.503, p < 0.001). Additionally, neuropsychological test performance showed significant correlations with T1 globus pallidus values: NCT-A scores demonstrated a moderate negative correlation (r = −0.345, 95% CI: −0.517 to −0.146, p < 0.001). DST scores showed a moderate positive correlation (r = 0.232, 95% CI: 0.024 to 0.421, p = 0.029) (Fig 3).

## Diagnostic performance of T1 and T2 values in MHE

ROC analysis revealed excellent diagnostic performance of T1 globus pallidus values for distinguishing between MHE and non-MHE patients with the AUC of 0.92 (95% CI: 0.86–0.99). The T1 globus pallidus values demonstrated a sensitivity of

 

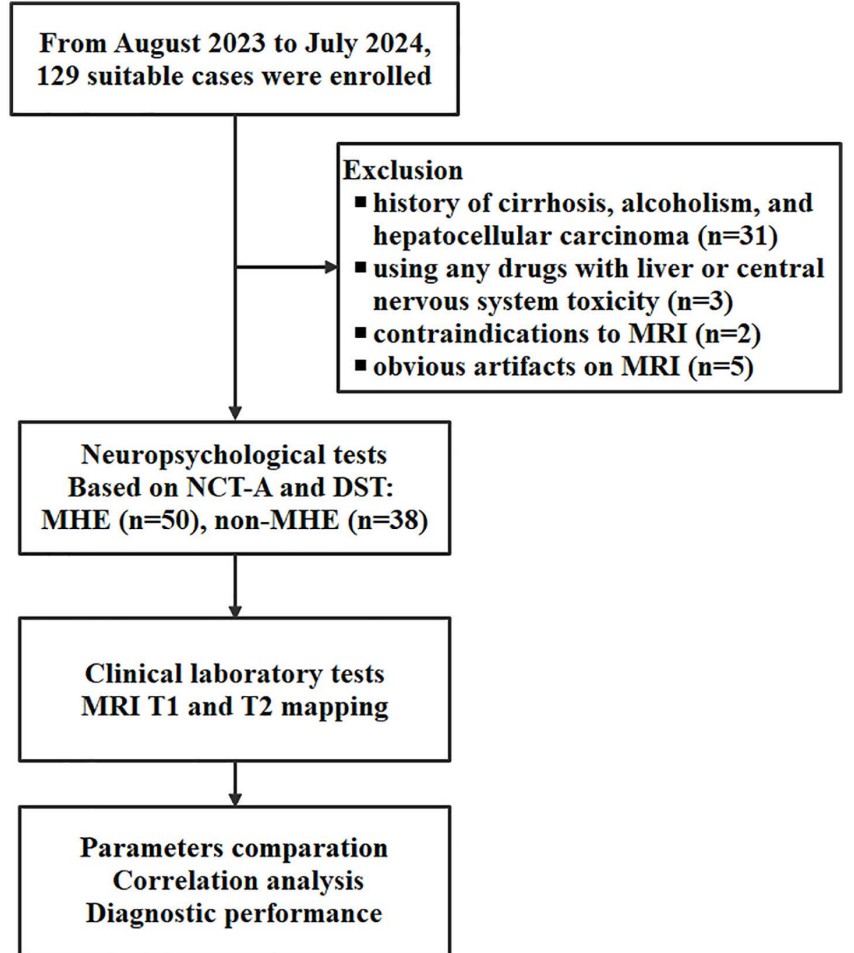

**Fig 1. The work flow of this study.** NCT-A, number connection test-A; DST, digit symbol test; MHE, minimal hepatic encephalopathy.

0.97 for detecting non-MHE patients and a specificity of 0.90 for excluding MHE cases with a PPV of 0.88, while the NPV was 0.98 (Table 3, Fig 4). In contrast, T2 values showed no good discriminative ability in any examined brain region (AUC range: 0.49–0.61).

## Discussion

This study demonstrated that the MRI T1 mapping can be used as a non-invasive diagnostic approach for detecting MHE in CHS patients. The results revealed that CHS-related MHE patients exhibited lower T1 values across multiple brain regions, particularly in the globus pallidus, whereas T2 mapping showed no significant diagnostic value. This study similarly identified T1 shortening as a reflection of manganese deposition secondary to portosystemic shunting.

Brain manganese deposition is a known phenomenon due to portosystemic shunting, which may serve as a sensitive biomarker for early neurological impairment in HE patients [13]. For the HE patients, the typical abnormalities observed in MR imaging the hyperintensities on T1-weighted scans, particularly in the globus pallidus due to manganese deposition [14]. Results of this study are consistent with findings in cirrhosis-related HE [15]. The strong negative correlation between globus pallidus T1 values and MHE suggests that manganese deposition in this region. However, research has indicated

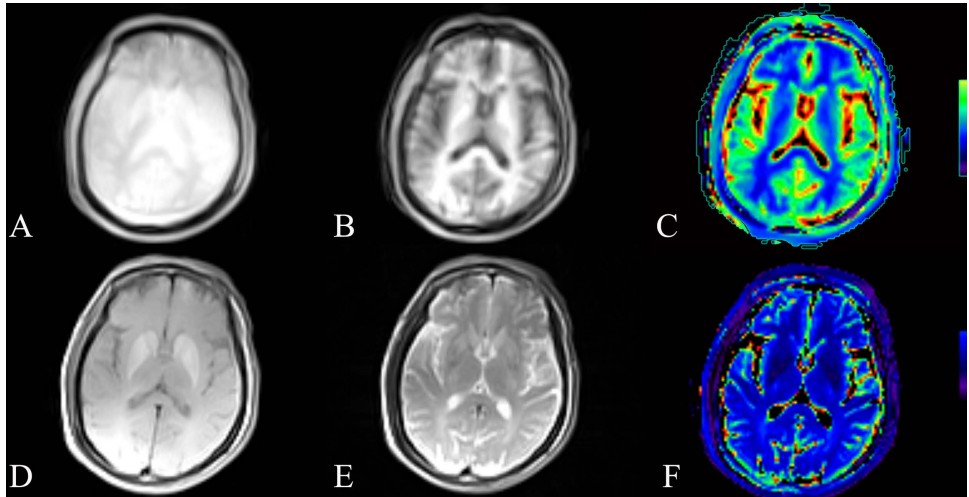

**Fig 2. T1 and T2 mapping in brain MRI. (A)** T1-weighted image with repetition time of 200 ms. **(B)** T1-weighted image with repetition time of 800 ms. **(C)** Quantitative T1 relaxation time map displayed in pseudo-color, where the color scale represents T1 relaxation times (blue indicating shorter T1 times, progressing through green and yellow to red indicating longer T1 times). **(D)** T2-weighted image with echo time of 10 ms. **(E)** T2-weighted image with echo time of 80 ms. **(F)** Quantitative T2 relaxation time map in pseudo-color format, with the color scale representing T2 relaxation times (purple representing shorter T2 times, transitioning through dark blue to light blue representing longer T2 times).

**Table 2. Result of the multivariable logistic regression analysis.**

| Parameters | Odd ratio | P-value | Conf low | Conf high |
|---|---|---|---|---|
| (Intercept) | 2.97 | 0.061 | 0.967 | 9.140 |
| Gender | 0.88 | 0.134 | 0.749 | 1.040 |
| Age | 1.00 | 0.372 | 0.984 | 1.010 |
| Education level | 0.86 | 0.250 | 0.659 | 1.110 |
| AST | 1.00 | 0.066 | 0.992 | 1.000 |
| T1 Frontal Lobe | 1.00 | 0.865 | 1.000 | 1.000 |
| T1 Temporal Lobe | 1.00 | 0.682 | 1.000 | 1.000 |
| T1 Occipital Lobe | 1.00 | 0.841 | 1.000 | 1.000 |
| T1 Caudate Nucleus | 1.00 | 0.055 | 1.000 | 1.000 |
| T1 Globus Pallidus | 1.00 | 0.004 | 1.000 | 1.000 |

that conventional T1-weighted hyperintensities may not accurately reflect clinical severity [16]. This limitation stems from the qualitative nature of standard T1-weighted imaging, which depends on radiologists' subjective evaluation of signal intensity [14].

In contrast, quantitative T1 mapping enables objective and reproducible measurement of T1 relaxation times [15]. However, previous studies showed that quantitative T1 mapping enables earlier and more reproducible detection of cirrhosis-related MHE and provides objective and reproducible measurements that can detect subclinical brain changes before overt neurological symptoms appear [17,18]. The present findings extend this knowledge to type B CHS-related MHE. The correlation between T1 values and psychometric test performance supports the idea that quantitative T1 mapping reflects early neurotoxic effects of manganese, even in the absence of overt hepatic dysfunction.

In contrast, T2 mapping showed no significant group differences. T2 value was previously reported with different manifestations in different brain regions of MHE patients. Elevated T2 values were reported in cirrhosis-related MHE patients

## Associations between MRI and clinical features

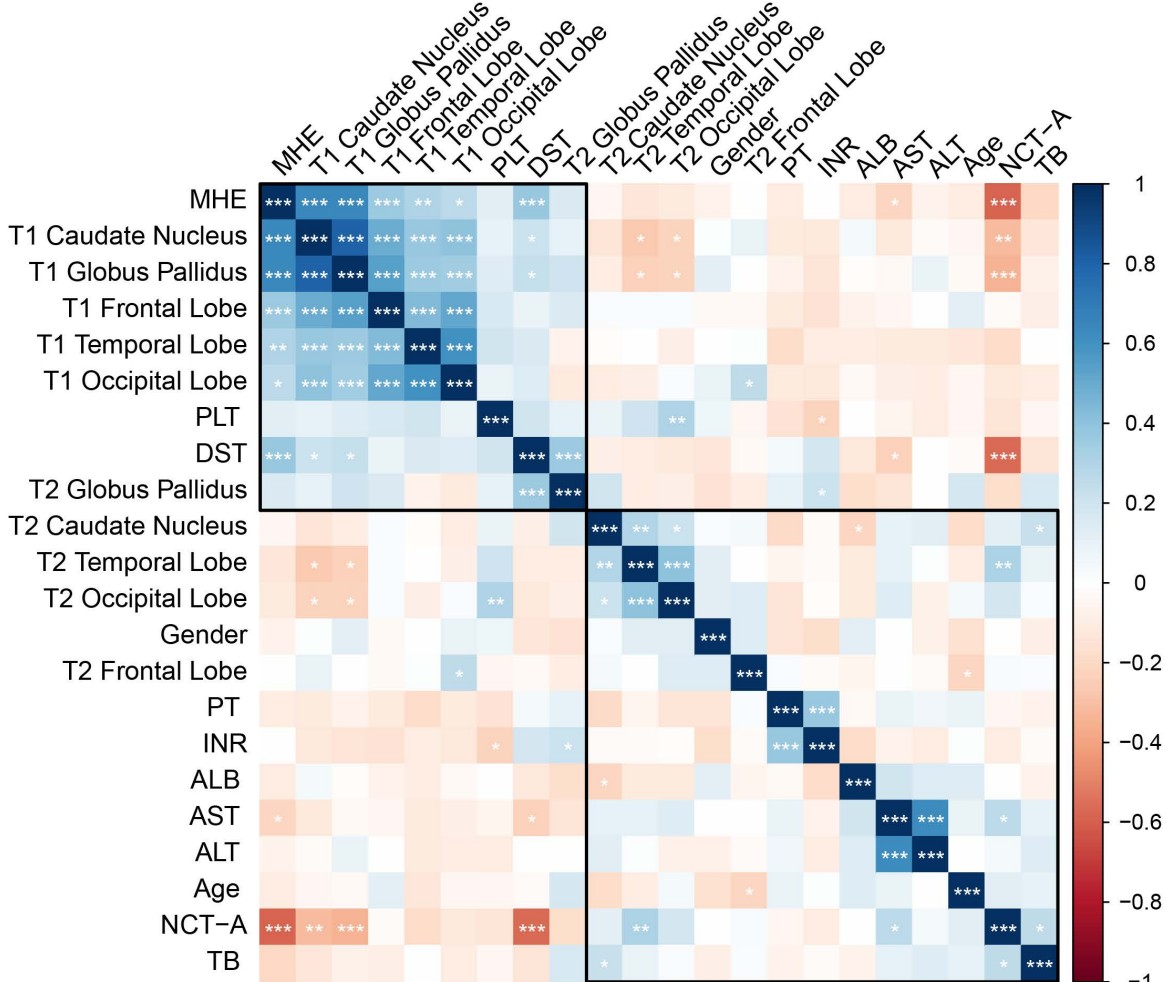

**Fig 3. Associations between MRI and clinical features.** Correlation matrix showing Pearson correlation coefficients between MRI-derived T1 and T2 values in different brain regions and clinical variables. The color scale represents correlation strength from −1 (red, strong negative correlation) to +1 (blue, strong positive correlation). Asterisks indicate statistical significance levels: * $p < 0.05$, ** $p < 0.01$, *** $p < 0.001$. ALB, albumin; ALT, alanine amino-transferase; AST, aspartate aminotransferase; INR, international normalized ratio; PLT, platelet count; PT, prothrombin time; TB, total bilirubin.

in frontal and occipital lobes, indicating increased free water content and low-grade brain edema due to hyperammonemia and dysfunction of astrocytes [19,20]. However, reduced T2 phase values within the frontal cortical-basal ganglia circuits was also reported in chirosis-related MHE patients, which could be the result of manganese disposition [21,22]. Our lack of T2 alteration suggests that in CHS-related MHE, T2 value is influenced by both brain edema and manganese deposition.

The neuropsychological findings in this study further support the clinical relevance of T1 mapping. Poorer performance on NCT-A and DST is correlated significantly with lower T1 values in the globus pallidus, which underscores the functional consequences of these imaging changes in the brain. Given the limitations of psychometric testing,

**Table 3. Performance of the T1 values for MHE in CHS patients.**

| Parameters | AUC | 95%CI | SPE | SEN | NPV | PPV |
|---|---|---|---|---|---|---|
| T1 Frontal Lobe | 0.73 | 0.62-0.84 | 0.64 | 0.76 | 0.78 | 0.62 |
| T1 Temporal Lobe | 0.69 | 0.58-0.8 | 0.42 | 0.92 | 0.88 | 0.55 |
| T1 Occipital Lobe | 0.70 | 0.59-0.81 | 0.68 | 0.68 | 0.74 | 0.62 |
| T1 Caudate Nucleus | 0.89 | 0.82-0.96 | 0.92 | 0.74 | 0.82 | 0.88 |
| T1 Globus Pallidus | 0.92 | 0.86-0.99 | 0.90 | 0.97 | 0.98 | 0.88 |

AUC, area under the receiver operating characteristic curve; CI, confidence interval; NPV, negative predictive value; PPV, positive predictive value; SEN, sensitivity; SPE, specificity. *P-values represent DeLong test comparisons between each model and the nomogram.

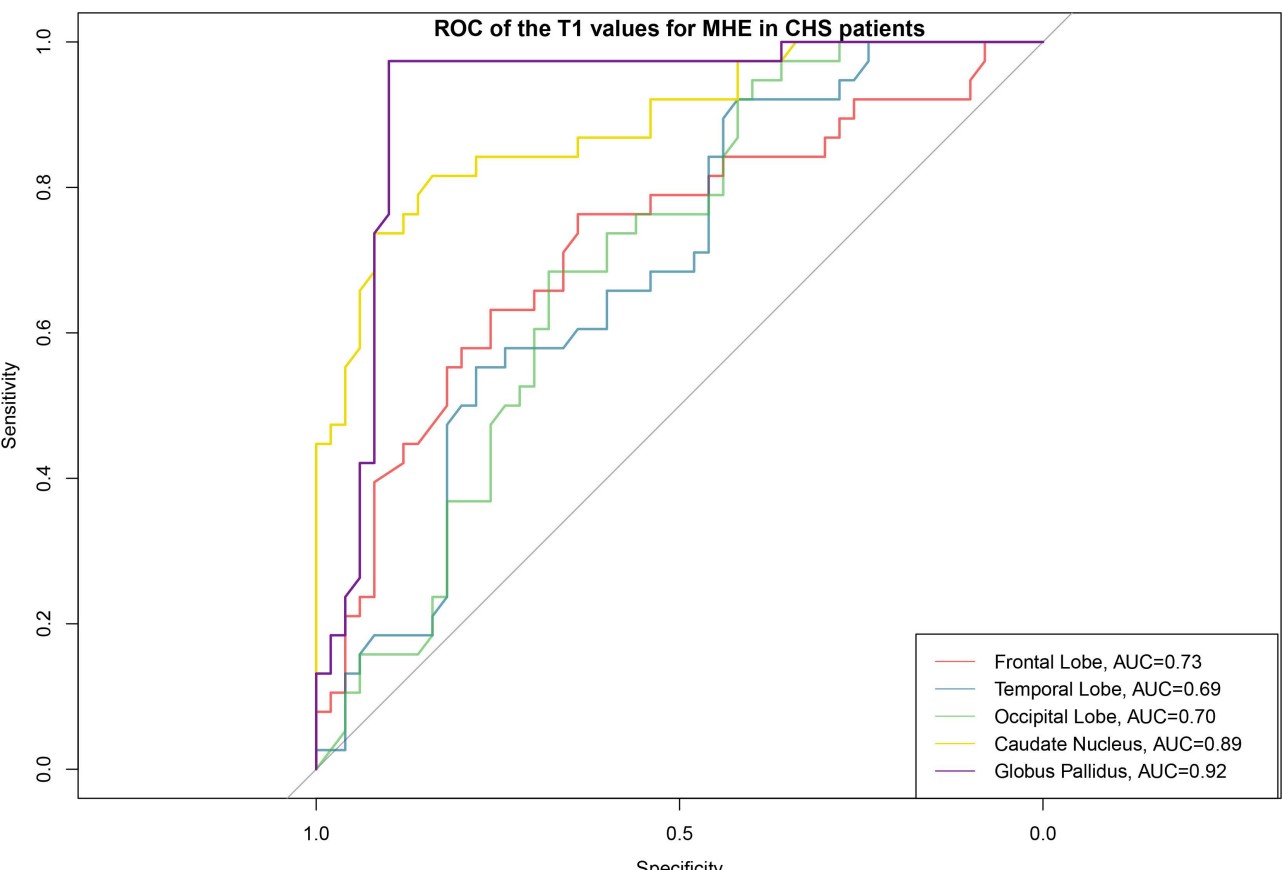

**Fig 4. Area under the receiver operating characteristic (ROC) curve (AUC) analysis of the T1 values for MHE in CHS patients.** The red, blue, green, yellow, and purple curves represent T1 Frontal Lobe, T1 Temporal Lobe, T1 Occipital Lobe, T1 Caudate Nucleus, and T1 Globus Pallidus, respectively.

including its time-consuming nature and sensitivity to education level, T1 mapping could provide a more efficient and objective method for identifying MHE in CHS patients [23]. Furthermore, the results suggest that T1 mapping might be incorporated into diagnostic protocols, particularly in resource-limited settings where specialized neuropsychological testing is unavailable [24].

This study has several strengths. First, it is the first to evaluate the diagnostic value of quantitative T1 and T2 mapping for CHS-related MHE. Second, the inclusion of psychometric testing and quantitative imaging correlations provides both functional and structural validation of the findings. Finally, by demonstrating the potential of quantitative T1 mapping as a non-invasive biomarker, this work suggests more objective and accessible diagnostic approaches in patients with CHS-related MHE.

However, some limitations should be acknowledged. First, although adequate for statistical analysis, the sample size was relatively small and drawn from a single center, which may limit the generalizability of the findings. And this was a cross-sectional study, preventing assessment of longitudinal changes in T1 values and their predictive value for clinical outcomes. Second, while T1 shortening is suggestive of manganese deposition, histopathological confirmation was not possible in our cohort. Third, compared with psychometric evaluations, the substantial expenses associated with MRI render its cost-effectiveness ratio potentially unappealing in areas with limited resources where schistosomiasis is prevalent. Hence, T1 mapping could be incorporated into a multimodal diagnostic strategy rather than employed as an independent screening assessment. Fourth, the elderly nature of our cohort is a limitation. It remains unclear if T1 mapping thresholds require age-specific calibration to maintain accuracy. Finally, the reliance on only the NCT-A and DST is a limitation, as it may slightly decrease sensitivity to subtle cognitive deficits compared to the full PHES.

## Conclusions

In conclusion, our study provides evidence that quantitative T1 mapping, especially in the globus pallidus, could be used as a potential tool for diagnosing MHE in CHS patients. Future longitudinal and multi-center studies are warranted to validate these findings and explore the role of T1 mapping in monitoring disease progression and guiding therapy.

## Supporting information

**S1 Table. The clinical and laboratory characteristics and T1 and T2 values in MHE and non-MHE cases.**
(DOCX)

## Author contributions

**Conceptualization:** Xin Li.

**Data curation:** Xue-Fei Liu, Ke-Ying Wang, Hai-Feng Shi, Ying Li.

**Formal analysis:** Xue-Fei Liu, Ying Li.

**Funding acquisition:** Xin Li.

**Writing – original draft:** Xue-Fei Liu, Ke-Ying Wang, Hai-Feng Shi, Ying Li, Xin Li.

**Writing – review & editing:** Ying Li, Xin Li.

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
