## [Decision Letter · Decision Letter 0]

13 Oct 2025

Dear Dr. Li,

Thank you for submitting your manuscript to PLOS ONE. After careful consideration, we feel that it has merit but does not fully meet PLOS ONE’s publication criteria as it currently stands. Therefore, we invite you to submit a revised version of the manuscript that addresses the points raised during the review process.

We look forward to receiving your revised manuscript.

Kind regards,

Ghada Abdrabo Abdellatif Elshaarawy, M.D

Academic Editor

PLOS ONE

4. Thank you for stating the following in the Funding Section of your manuscript:

“This work was supported by grant No. 2023-WS-07 to investigator Xin Li from Shanghai Jinshan District Science and Technology Committee.”

Additional Editor Comments:

1) ABSTRACT:

The abstract will be better if structured in this way: background, aim, methods, results, and conclusion.

Methods is lacking the study setting.

2) INTRODUCTION:

Newest Global/ Regional/ China prevalence of minimal hepatic encephalopathy among chronic hepatic schistosomiasis patients should be stated.

The current situation of other developed and developing countries should also be added with new references.

Explaining why this topic was chosen for analysis in this article is not well written. The benefits of conducting the study to the community should be explained.

3) METHODS:

How did the authors get the study subjects? They have to clearly address their sampling technique?

The study design was a cross-sectional observational study not a forward-looking observational study.

Methods is lacking the study setting.

How the Authors got the socioeconomic data from the patients?

How the participants answered the questionnaire, by themselves or with assistance from researcher?

How long did it take to complete each test/questionnaire?

The authors should mention the criteria of the study setting.

It is advisable to include the questions as per each domain in the methodology.

How to score the NCT-A and the DST tests, specify the cut points?

T1 and T2 mapping need more clarification of their indication and usage.

What is the difference between conventional T1-weighted imaging and quantitative T1 mapping?

What strategy was devised to increase the accuracy of the study and the accuracy of the answers? What types of technique the authors used to keep/control the data quality?

Authors should include a reference for using the stated formula in calculating the sample size. Furthermore, the basis of sample size calculation should be mentioned to know the confidence level and the margin of error. How did you determine the sample size? Please give a justification for the sample size.

4) RESULTS:

Number of the study participants in both groups in Figure 1. The work flow of this study was differed from what written in the result’s comment.

The result section is lacking tables for NCT-A and DST scores, multivariable logistic regression, and ROC analysis figure and table.

Multivariable logistic regression table should contain all variables enter in the regression.

5) DISCUSSION:

The authors had not done a good job of connecting their findings to the literature and interpreting their findings. Discuss by using the scientific reasoning the MRI T1 and T2 mapping for detecting MHE among CHS patients in other developing and developed countries with similar context. The manuscript could be greatly strengthened if the authors could compare the findings of the study with other findings and state the reasons for the strengths and weaknesses in each section.

6) STRENGTHS:

Please analyze the strengths of the study.

Reviewers' comments:

Reviewer's Responses to Questions

**Comments to the Author**

1. Is the manuscript technically sound, and do the data support the conclusions?

Reviewer #1: Partly

Reviewer #2: Yes

Reviewer #3: Yes

Reviewer #4: Yes

Reviewer #5: Yes

2. Has the statistical analysis been performed appropriately and rigorously?

Reviewer #1: No

Reviewer #2: Yes

Reviewer #3: Yes

Reviewer #4: Yes

Reviewer #5: Yes

3. Have the authors made all data underlying the findings in their manuscript fully available?

Reviewer #1: Yes

Reviewer #2: Yes

Reviewer #3: Yes

Reviewer #4: Yes

Reviewer #5: Yes

4. Is the manuscript presented in an intelligible fashion and written in standard English?

Reviewer #1: No

Reviewer #2: Yes

Reviewer #3: Yes

Reviewer #4: Yes

Reviewer #5: Yes

Reviewer #1: Peer Review

General comments

The manuscript investigates whether quantitative T1 and T2 mapping can serve as non-invasive biomarkers for minimal hepatic encephalopathy (MHE) in patients with chronic hepatic schistosomiasis (CHS). The study addresses a relevant knowledge gap, as MRI has no established diagnostic role in MHE and is mainly a research tool. The design is prospective in recruitment but cross-sectional in execution, as all measurements were taken at a single time point. The manuscript is overall well organized, clearly written, and the English is of generally good quality, though a few sentences could be streamlined for conciseness. Several aspects of the introduction, methods, results, and discussion require clarification or refinement to strengthen transparency and reproducibility.

Introduction

1. The introduction provides useful background but is somewhat lengthy and diffuse. The rationale could be sharpened to help the reader follow clearly what is new, what is relevant, and how MRI is currently used. At present, MRI findings such as T1 hyperintensities are research observations reflecting manganese deposition but are not part of routine diagnostic criteria for MHE and do not replace psychometric testing in clinical practice. Quantitative T1 mapping has shown promise in cirrhosis and animal models but has not previously been studied in CHS. Making this distinction more explicit would clarify the knowledge gap and highlight the novelty of the present study.

2. The final paragraph should concisely state the study’s objective and hypothesis (e.g., that quantitative T1 mapping, especially in the globus pallidus, may discriminate between CHS patients with and without MHE).

Methods

3. The study design should be described as a cross-sectional observational study, rather than “forward-looking observational.”

4. The sample size calculation is reported, which is a strength, but the assumptions are unclear. A reference or justification for the expected effect size should be provided, and it should be specified whether the same calculation applied to both T1 and T2 outcomes.

5. Exclusion criteria include “history of cirrhosis.” Since schistosomiasis patients may have periportal fibrosis or secondary portal hypertension, the operational definition of cirrhosis and how it was excluded should be clarified.

6. Diagnosis of MHE relied on NCT-A and DST. The authors should justify this choice, since PHES is considered the gold standard internationally. It would also be useful to discuss how much diagnostic overlap exists with PHES and whether using only two tests may reduce accuracy.

7. The MRI protocol is described in detail, but further clarification is needed: were regions analyzed automatically or manually? If manual, how many raters were involved, and was inter-rater reliability (e.g., ICC) calculated? Were image readers blinded to test results or group allocation?

8. Blinding of neuropsychological test assessors should also be stated.

9. The regression models are not fully transparent. The covariates included should be explicitly listed, with justification for their selection. Handling of multiple comparisons should also be mentioned. F

10. Figure 1 illustrates the study workflow but does not report the number of patients screened, excluded, and included. For clarity and compliance with reporting standards, these numbers should be added.

Results

11. The results are clearly presented, but Table 1 contains many non-significant variables that could be moved to supplementary material or presented more concisely.

12. Confidence intervals should be consistently reported alongside p-values to improve interpretability.

13. The covariates used in regression should be specified in both methods and results to increase transparency.

Discussion

14. The very high AUC values (0.92 with sensitivity 0.97 and specificity 0.90) should be interpreted cautiously. Possible sources of overestimation, such as single-center recruitment, sample size, or limited cohort variability, should be discussed.

15. The statement that T1 mapping is “more objective and reproducible” than conventional T1-weighted imaging should be supported with references or rephrased more cautiously.

16. The limitations of psychometric testing related to education are important, but no socioeconomic or educational baseline variables are reported. If these data are unavailable, this should be acknowledged as a limitation.

17. Clinical implications should be framed carefully. At present, T1 mapping should be considered as a complementary tool rather than a replacement for psychometric testing. Barriers such as availability and cost should also be acknowledged.

Conclusion

18. The conclusion is consistent with the findings but should avoid overstatement. Instead of claiming “strong evidence,” the study should be described as providing promising evidence that T1 mapping has potential as a biomarker for MHE in CHS.

Final remarks

This is a promising and well-prepared study with clear potential to contribute to the field. With clarification of study design, diagnostic rationale, imaging analysis, blinding, covariate selection, and interpretation of diagnostic accuracy, the manuscript will be significantly strengthened.

Thank you for the opportunity to review this work!

Reviewer #2: Well written and articulated study. If you can elaborate if similar T1 and T2 findings seen in cirrhosis related MHE or if this is different readers will have a better understanding.

How were the candidates selected for the scan? If it was every single patient presenting to the clinic with CHS or was there a minimal severity required?

Reviewer #3: Strengths

Scientific rigor: The study design, inclusion/exclusion criteria, and sample size calculation are appropriate. Ethical approval and informed consent were obtained.

Statistical analysis: Statistical methods (group comparisons, logistic regression, correlation analyses, and ROC analysis) are well described and correctly applied. Results support the conclusions drawn.

Data availability: All underlying data have been made publicly accessible through the provided repository link, consistent with PLOS ONE data policies.

Clarity and language: The manuscript is clearly written in standard English, with logical organization and understandable figures and tables.

Minor Suggestions for Revision

Consistency in terms: Ensure consistent use of “CHS” vs. “CHE” (one instance of “CHE” appears in the Results section).

Overall, the manuscript is of high quality, scientifically robust, and suitable for publication after minor editorial refinements.

Reviewer #4: (Please, see the attachment for detailed comments and suggestions.)

The authors’ work significantly enhances the diagnostic utility of quantitative T1 and T2 mapping as an accurate and non-invasive biomarker for detecting MHE in CHS patients. The measurement of T1 and T2 values, as well as magnetic susceptibility values, in different brain regions could contribute to the recognition of early neurological symptoms and the monitoring of disease progression in patients with CHS. However, their findings reveal a strong correlation between T1 values and the globus pallidus in CHS patients with MHI, exhibiting higher sensitivity and specificity diagnostic magnetic values. Since the central theme of the research discussion centers on T1 mapping, manganese deposition, and magnetic susceptibility of the globus pallidus in differentiating MHI from non-MHI in CHS patients. The following fundamental points have to be brought to the attention of the authors:

There is discordance between the title of the manuscript and the findings of the research study. The title highlights equally the relevance and quantitative role of T1 and T2 mapping in detecting minimal hepatic encephalopathy in patients with chronic hepatic schistosomiasis, while 80% of the discussion prominently emphasizes T1 mapping. The introduction of the manuscript lacks background information and a review of previous work on T2 mapping.

The authors defined the aim of this study as evaluating whether MRI T1 mapping can be used for clinical diagnoses in MHE in CHS patients. The title should be more suitable if it focuses on T1 mapping because the analysis attempts to explore T2 mapping, which yields non-significant outcomes.

The physiological effects of age on mental function, the cognitive role of psychometric test outcomes, and natural variations in T1 and T2 values with aging should be considered in the methodological aspect and incorporated as part of the discussion section.

The mean age of the patients was approximately 70, and most were in their 5th, 6th, and 7th decades of life. The context of this research paper and the study's outcomes should be tailored to the quantitative role and diagnostic accuracy of T1 mapping in diagnosing MHE in middle-aged and older CHS patients.

The key message of this scientific study is that MRI findings characterized T1 hyperintensity in the globus pallidus of CHS in patients, with higher diagnostic accuracy in differentiating MHE from non-MHI in CHS patients with a mean age of 70 years. Furthermore, emphasizing its significance over conventional psychometric tests as a non-invasive and objective tool for detecting MHE in CHS patients, and potentially serving as a surrogate biomarker for the progression of chronic liver diseases.

Reviewer’s Comments and Suggestions

Please define the acronym “hepatic encephalopathy (PH)” upon its first mention, and use it consistently thereafter.

“ PHES is considered the “gold” standard diagnosis for MHE, which includes five tests.”

The sentence requires reference, and it would be informative to list down the tests briefly, accompanied with a reference as follows: “ The PHES (psychometric hepatic encephalopathy score) test is deemed the gold standard in the diagnosis of MHE, consisting of 5 paper-and-pencil test sheets: Number connection tests (NCT-A and NCT-B), line tracing test (LTT), digit symbol test(DST), and serial dotting test (SDT).

The authors mentioned that the diagnosis of MHE relies on the results of two abnormal tests. Which two tests are you referring to? The EASL/AASL 2014 Guidelines on Hepatic Encephalopathy recommended using at least two tests to detect MHE, based on resulting alterations in psychometric and neurophysiologic tests. However, the updated EASL guidelines from 2022 recommend relying on a single test, with an emphasis on available local norms and experience. Can you give charity on this matter?

The authors highlighted two significant limitations of applying PHES tests for diagnosing MHE: the time-consuming nature of the tests and the influence of participants' educational level on outcomes. However, the authors fall short in addressing the age of study subjects as one of the prominent factors correlated with mental function, performance, and patient cognitive decline.

According to (Randolph, C. et al.), the Neuropsychological Assessment of Hepatic Encephalopathy: ISHEN’s 2009 Practice Guidelines pointed out that the PHES test findings are impacted by various factors, including age, gender, level of education, alcohol consumption, and visual abilities of the individual attending the psychometric test. Since the test focuses on multiple domains (processing speed, working memory, mental attention, visual scanning efficacy and perception, fine motor function, and motor speed), the age of the test subject, as well as occupation, social background, and gender, affect the sensitivity of the tests and shouldn’t be left unnoted as statistically significant parameters of performance.

Several scientific studies stated that neuronal loss and manganese accumulation due to hepatic encephalopathy can lead to bilateral, reversibly symmetrical T1 hyperintensity in the globus pallidus and substantia nigra areas of the brain.

Evidence from Magnetic resonance imaging has shown that T2 hyperintensity occurs in the white matter along the corticospinal tract in the brains of patients with advanced hepatic encephalopathy and cirrhosis. Encephalopathy (MHE) does not have a consistent or specific MRI T2 hyperintensity signature, as MHE is primarily a clinical diagnosis based on psychometric tests. What is the rationale behind including T2 mapping in your research study?

In your ethical considerations section, you mentioned the potential risks and benefits involved in the study. Are you referring to the risks and benefits of MRI examination? Or, the rewards and drawbacks of T1 and T2 mapping? Can you elaborate on it with one or two lines?

What are you implying by using the term forward-looking observational study? Was it a retrospective study with follow-up measurements of previously collected data from patients? Or a prospective longitudinal survey observing the two cohorts for a specific period of time?

The study design stated that the participants were divided into two respective groups (with MHI and without MHI), with a sample size of approximately 36 patients. However, your study flow diagram (Fig. 1) reveals a discrepancy in sample size between these two groups (52 vs. 36). Could you provide clarity on this matter?

Were the 50 patients with MHE and 38 patients without MHE (non-MHE) comparable groups in your study?

Your exclusion criteria were a history of cirrhosis, alcoholism, hepatocellular carcinoma, and medication usage. Did you have a patient's information on historical neuropsychological disorders, brain injury, and neurological diseases that potentially affect the cognitive function and bias the assessment of study subjects?

Was magnetic resonance imaging employed in the heart or brain regions? Can you provide us with the context for why your MRI examinations were specifically targeted to assess myocardial tissue characterization? Or, is it a typographic error that needs correction?

What normality test did you use to assess the distribution of the continuous variable?

Did you have any missing values and outliers in your variables? How did you handle it? What techniques were implemented during the data cleaning phases?

Fig. 3 shows the association between MRI and clinical features using a Pearson correlation matrix. Did you use the Spearman correlation matrix in your univariate analysis? Since you stated Pearson and Spearman correlation analyses in your statistical section, which variables were found to be non-normally distributed? Please specify the findings in your results section.

Multivariate adjustment is reported in your results section. Please specify all your analysis techniques clearly in the statistical section.

What variables were adjusted in your logistic regression? Define your dependent variable clearly, independent predictors, and adjusted co-founders in your model.

The authors aimed to evaluate the diagnostic accuracy of T1 and T2 mapping for detecting MHE. However, the diagnostic performance of T2 is missing in the results section. Did your T2 mapping comparison between groups show statistically non-significant results in all tests, including descriptive, univariate, multivariate, and ROC analysis? If that is the case, the scope of your research study should focus on only T1 mapping rather than T2.

N (%). Please specify it in the footnotes of Table 1 that your continuous variables are presented as the mean ± standard deviation, and categorical variables are presented as frequency values in numbers and percentages (%).

It would be informative if you could present multivariate analysis as Table 2 and ROC analysis as Fig. 3.

Reviewer #5: Methods and Results

Did you identify any confounders in this study?

What parameters were adjusted for in this study?

Include results of the sensitivity analysis done to ensure robustness of the data presented

**Do you want your identity to be public for this peer review?** For information about this choice, including consent withdrawal, please see our Privacy Policy

Reviewer #1: No

Reviewer #2: **Yes: ** Avinash Govinda Adiga

Reviewer #3: **Yes: ** Hala Awad Ahmed

Reviewer #4: **Yes: ** Robel A. Habte

Reviewer #5: **Yes: ** Sheneil Agyemang

---

## [Author Response · Author response to Decision Letter 1]

20 Oct 2025

Thank you for your detailed review and constructive feedback. We've carefully addressed each comment with comprehensive revisions. Below is a point-by-point response, with changes highlighted in bold or tracked in the attached document for clarity.

Additional Editor Comments:

1) ABSTRACT:

The abstract will be better if structured in this way: background, aim, methods, results, and conclusion.

Response: Thank you for the suggestion. We have structured the abstract as suggested.

Methods is lacking the study setting.

Response:

Thank you for the comment. We have revised the Methods of the abstract to include the study setting.

2) INTRODUCTION:

Newest Global/ Regional/ China prevalence of minimal hepatic encephalopathy among chronic hepatic schistosomiasis patients should be stated.

The current situation of other developed and developing countries should also be added with new references.

Explaining why this topic was chosen for analysis in this article is not well written. The benefits of conducting the study to the community should be explained.

Response: Thank you for the suggestions. We have revised the introduction to include updated prevalence data, situation of other developed and developing countries, and a clearer rationale for the study.

3) METHODS:

How did the authors get the study subjects? They have to clearly address their sampling technique?

Response: Thank you for the comment. We have revised the manuscript to provide explicit details about our sampling technique (Materials and methods-Data acquisition).

The study design was a cross-sectional observational study not a forward-looking observational study.

Response: Thank you for the reminder. This is a cross-sectional observational study. We corrected it in the text (Materials and methods-Study design and determination of sample size).

Methods is lacking the study setting.

How the Authors got the socioeconomic data from the patients?

Response: Thank you for the comment. In this study, we did not collect detailed socioeconomic variables such as income or occupation. However, we recorded education level along with medical history and age, which were obtained through a structured questionnaire completed during patient enrollment, before neuropsychological testing (Materials and methods-Data acquisition).

How the participants answered the questionnaire, by themselves or with assistance from researcher?

How long did it take to complete each test/questionnaire?

The authors should mention the criteria of the study setting.

It is advisable to include the questions as per each domain in the methodology.

How to score the NCT-A and the DST tests, specify the cut points?

Response: Thank you for the comment. We have added details regarding the test administration procedure, completion times for each assessment, details about the study setting (Materials and methods-Assessment of minimal hepatic encephalopathy).

T1 and T2 mapping need more clarification of their indication and usage.

Response: Thank you for the comment. We have revised the manuscript to provide a clearer explanation of T1 and T2 mapping methodologies and their specific clinical applications (Materials and methods-Magnetic resonance imaging protocol).

What is the difference between conventional T1-weighted imaging and quantitative T1 mapping?

Response: Thank you for the inquiry. Conventional T1-Weighted Imaging provides a qualitative method based on signal intensity, relying on radiologists' subjective interpretation. Quantitative T1 Mapping offers objective, reproducible assessment of T1 relaxation times.

What strategy was devised to increase the accuracy of the study and the accuracy of the answers? What types of technique the authors used to keep/control the data quality?

Response: Thank you for the inquiry. All MRI examinations and psychometric tests followed standardized operating protocols. Two experienced radiologists, blinded to clinical data, independently measured T1 and T2 values, with discrepancies resolved by consensus. Images with artifacts were excluded from analysis. Psychometric tests were administered by trained examiners using uniform instructions. Data entry was performed in duplicate and cross-checked to minimize transcription errors.

Authors should include a reference for using the stated formula in calculating the sample size. Furthermore, the basis of sample size calculation should be mentioned to know the confidence level and the margin of error. How did you determine the sample size? Please give a justification for the sample size.

Response: Thank you for the comment. We revised the text to clarify the sample size calculating method (Materials and methods-Study design and determination of sample size).

4) RESULTS:

Number of the study participants in both groups in Figure 1. The work flow of this study was differed from what written in the result's comment.

Response: Thank you for the comment. We have included the number of the study participants and modified the work flow to be consistent with the text.

The result section is lacking tables for NCT-A and DST scores, multivariable logistic regression, and ROC analysis figure and table.

Response: Thank you for the comment. The NCT-A and DST scores have been included in Table 1. The result of the multivariable logistic regression has been included in Supplementary Table 2. ROC analysis has been included in the Table 2.

Multivariable logistic regression table should contain all variables enter in the regression.

Response: Thank you for the comment. We considered age, gender, and education level as potential confounding variables, along with the variables with p < 0.05 (AST, T1 frontal lobe, T1 temporal lobe, T1 occipital lobe, T1 caudate nucleus, and T1 globus pallidus) in univariate analyses. The result of the multivariable logistic regression analysis is now provided in the Supplementary Table 2.

5) DISCUSSION:

The authors had not done a good job of connecting their findings to the literature and interpreting their findings. Discuss by using the scientific reasoning the MRI T1 and T2 mapping for detecting MHE among CHS patients in other developing and developed countries with similar context. The manuscript could be greatly strengthened if the authors could compare the findings of the study with other findings and state the reasons for the strengths and weaknesses in each section.

Response: Thank you for the suggestion. We revised the Discussion to highlight the strengths and limitations of our results relative to other studies.

6) STRENGTHS:

Please analyze the strengths of the study.

Response: Thank you for the suggestion. We added a paragraph summarizing the key methodological and clinical strengths of this study.

Reviewer #1: Peer Review

Introduction

1.The introduction provides useful background but is somewhat lengthy and diffuse. The rationale could be sharpened to help the reader follow clearly what is new, what is relevant, and how MRI is currently used. At present, MRI findings such as T1 hyperintensities are research observations reflecting manganese deposition but are not part of routine diagnostic criteria for MHE and do not replace psychometric testing in clinical practice. Quantitative T1 mapping has shown promise in cirrhosis and animal models but has not previously been studied in CHS. Making this distinction more explicit would clarify the knowledge gap and highlight the novelty of the present study.

Response: Thank you for the suggestion. We have revised the introduction as suggested.

2. The final paragraph should concisely state the study's objective and hypothesis (e.g., that quantitative T1 mapping, especially in the globus pallidus, may discriminate between CHS patients with and without MHE).

Response: Thank you for the suggestion. We revised the introduction to make the study's objective and hypothesis clearer for the readers (Introduction-last paragraph).

Methods

3. The study design should be described as a cross-sectional observational study, rather than "forward-looking observational."

Response: Thank you for the reminder. This is a cross-sectional observational study. We corrected it in the text (Materials and methods-Study design and determination of sample size).

4. The sample size calculation is reported, which is a strength, but the assumptions are unclear. A reference or justification for the expected effect size should be provided, and it should be specified whether the same calculation applied to both T1 and T2 outcomes.

Response: Thank you for the comment. We revised the text to clarify the sample size calculating method (Materials and methods-Study design and determination of sample size).

5. Exclusion criteria include "history of cirrhosis." Since schistosomiasis patients may have periportal fibrosis or secondary portal hypertension, the operational definition of cirrhosis and how it was excluded should be clarified.

Response: Thank you for the comment. Cirrhosis was defined based on a combination of clinical history. Patients with a history of cirrhosis were excluded.

6. Diagnosis of MHE relied on NCT-A and DST. The authors should justify this choice, since PHES is considered the gold standard internationally. It would also be useful to discuss how much diagnostic overlap exists with PHES and whether using only two tests may reduce accuracy.

Response: Thank you for the comment. We have added text to clarify this justification and briefly discuss potential diagnostic limitations (Discussion-last paragraph).

7. The MRI protocol is described in detail, but further clarification is needed: were regions analyzed automatically or manually? If manual, how many raters were involved, and was inter-rater reliability (e.g., ICC) calculated? Were image readers blinded to test results or group allocation?

Response: Thank you for the comment. Quantitative T1 and T2 values were obtained by manually delineating regions of interest (ROIs) in the frontal, temporal, and occipital lobes, caudate nucleus, and globus pallidus. Two experienced radiologists, blinded to clinical data, independently measured T1 and T2 values, with discrepancies resolved by consensus. The mean of the two measurements was used for statistical analysis (Materials and methods-Magnetic resonance imaging protocol). Although the inter-rater reliability (ICC) was not calculated, both radiologists followed a standardized ROI placement protocol to ensure consistency across measurements.

8. Blinding of neuropsychological test assessors should also be stated.

Response: Thank you for the comment. We mentioned it in the text (Materials and methods-Magnetic resonance imaging protocol).

9. The regression models are not fully transparent. The covariates included should be explicitly listed, with justification for their selection. Handling of multiple comparisons should also be mentioned.

Response: Thank you for the comment. We considered age, gender, and education level as potential confounding variables, along with the variables with p < 0.05 (AST, T1 frontal lobe, T1 temporal lobe, T1 occipital lobe, T1 caudate nucleus, and T1 globus pallidus) in univariate analyses. The result of the multivariable logistic regression analysis is now provided in the Supplementary Table 2.

10. Figure 1 illustrates the study workflow but does not report the number of patients screened, excluded, and included. For clarity and compliance with reporting standards, these numbers should be added.

Response: Thank you for the comment. We have included the number of the study participants and modified the work flow to be consistent with the text.

Results

11. The results are clearly presented, but Table 1 contains many non-significant variables that could be moved to supplementary material or presented more concisely.

Response: Thank you for the suggestion. We have moved the non-significant variables to the Supplementary Table 1.

12. Confidence intervals should be consistently reported alongside p-values to improve interpretability.

Response: Thank you for the suggestion. We have revised the manuscript to include 95% confidence intervals for mean differences, odds ratios, and other relevant estimates in the Results section and in Supplementary Table 1.

13. The covariates used in regression should be specified in both methods and results to increase transparency.

Response: Thank you for the suggestion. We have added the methods and results of the multivariable logistic regression analysis in the text (Statistical Analysis and Multivariable adjustment and correlation analyses).

Discussion

14. The very high AUC values (0.92 with sensitivity 0.97 and specificity 0.90) should be interpreted cautiously. Possible sources of overestimation, such as single-center recruitment, sample size, or limited cohort variability, should be discussed.

Response: Thank you for the suggestion. We have revised the manuscript to discuss these limitations (Discussion-last paragraph).

15. The statement that T1 mapping is "more objective and reproducible" than conventional T1-weighted imaging should be supported with references or rephrased more cautiously.

Response: Thank you for the suggestion. We have revised the manuscript with more appropriate language (Conclusions).

16. The limitations of psychometric testing related to education are important, but no socioeconomic or educational baseline variables are reported. If these data are unavailable, this should be acknowledged as a limitation.

Response: Thank you for the suggestion. We added the information about educational level in the text.

17. Clinical implications should be framed carefully. At present, T1 mapping should be considered as a complementary tool rather than a replacement for psychometric testing. Barriers such as availability and cost should also be acknowledged.

Response: Thank you for the suggestion. We have revised the conclusion and discussion as suggested.

Conclusion

18. The conclusion is consistent with the findings but should avoid overstatement. Instead of claiming "strong evidence," the study should be described as providing promising evidence that T1 mapping has potential as a biomarker for MHE in CHS.

Response: Thank you for the suggestion. We revised the conclusion in both the abstract and the main text to avoid overstatement about the clinical usefulness of quantitative T1 mapping.

Reviewer #2: Well written and articulated study. If you can elaborate if similar T1 and T2 findings seen in cirrhosis related MHE or if this is different readers will have a better understanding.

Response: Thank you for the suggestion. We have revised the manuscript to include a comparative discussion of these findings (Discussion).

How were the candidates selected for the scan? If it was every single patient presenting to the clinic with CHS or was there a minimal severity required?

Response: Thank you for the inquiry. In our study, consecutive CHS patients referred to the local hepatology clinics and the gastroenterology department were screened for eligibility. All patients were invited to participate if they had a confirmed diagnosis of CHS and were willing and able to undergo MRI examination and psychometric testing. The MHE status was determined after enrollment based on psychometric test results (NCT-A and DST), allowing classification into MHE and non-MHE groups for comparative analysis.

Reviewer #3: Strengths

Minor Suggestions for Revision

Consistency in terms: Ensure consistent use of "CHS" vs. "CHE" (one instance of "CHE" appears in the Results section).

Response: Thank you for the suggestion. We revised CHE to CHS for consistency throughout the text.

Reviewer #4:

Reviewer's Comments and Suggestions

Please define the acronym "hepatic encephalopathy (PH)" upon its first mention, and use it consistently thereafter.

Response: Thank you for the suggestion. We have used "HE" as the acronym for hepatic encephalopathy and used it consistently throughout the text.

" PHES is considered the "gold" standard diagnosis for MHE, which includes five tests."

The sentence requires reference, and it would be informative to list down the tests briefly, accompanied with a reference a

---

## [Decision Letter · Decision Letter 1]

16 Nov 2025

Dear Dr. Li,

Thank you for submitting your manuscript to PLOS ONE. After careful consideration, we feel that it has merit but does not fully meet PLOS ONE’s publication criteria as it currently stands. Therefore, we invite you to submit a revised version of the manuscript that addresses the points raised during the review process.

We look forward to receiving your revised manuscript.

Kind regards,

Ghada Abdrabo Abdellatif Elshaarawy, M.D

Academic Editor

PLOS ONE

Journal Requirements:

Additional Editor Comments:

It is advisable to add the figure of the ROC curve analysis and the table of multivariable logistic regression in the result section rather than the supplementary materials.

Make sure you thoroughly address the reviewers’ comments and concerns and ensure the manuscript is free of any editorial or grammatical errors.

Reviewers' comments:

Reviewer's Responses to Questions

**Comments to the Author**

Reviewer #1: (No Response)

Reviewer #2: All comments have been addressed

Reviewer #3: All comments have been addressed

Reviewer #4: All comments have been addressed

2. Is the manuscript technically sound, and do the data support the conclusions?

Reviewer #1: Yes

Reviewer #2: Yes

Reviewer #3: Yes

Reviewer #4: Yes

3. Has the statistical analysis been performed appropriately and rigorously?

Reviewer #1: Yes

Reviewer #2: Yes

Reviewer #3: Yes

Reviewer #4: Yes

4. Have the authors made all data underlying the findings in their manuscript fully available?

Reviewer #1: Yes

Reviewer #2: Yes

Reviewer #3: Yes

Reviewer #4: Yes

5. Is the manuscript presented in an intelligible fashion and written in standard English?

Reviewer #1: Yes

Reviewer #2: Yes

Reviewer #3: Yes

Reviewer #4: Yes

Reviewer #1: Overall evaluation

The authors have carefully and comprehensively addressed the comments from the first review round. The revised manuscript is substantially improved, with clearer structure, better methodological transparency, and improved integration of supporting literature. The inclusion of detailed descriptions for sampling, test administration, and MRI analysis, as well as the revised figure showing participant flow, significantly enhances clarity and compliance with reporting standards. The English language is clear and appropriate for publication, although a brief copyedit before acceptance would further improve conciseness and fluency.

Remaining minor points for clarification

1. Blinding procedures

The revised text specifies that two radiologists independently analyzed MRI data blinded to clinical information, which is commendable. However, it remains unclear whether they were also blinded to participants' MHE classification status. I suppose they were, but this could be explicitly stated if it was the case. Similarly, while the psychometric tests were administered using standardized instructions by trained examiners, it is not explicitly stated whether these examiners were aware of the patients' cognitive or clinical background. Clarifying these aspects, whether both MRI readers and psychometric assessors were fully blinded to group status, would help the reader better assess the internal validity of the findings and minimize concerns about potential measurement bias.

2. Education variable

Including education level in Table 1 is an important improvement, but the variable ("above/below primary school") remains rather coarse. Education is known to influence psychometric performance through factors such as literacy, occupational complexity, and cognitive stimulation, which may not be captured by this binary measure. The discussion could benefit from a short acknowledgment that this simplified categorization may not fully account for the effect of educational background on test performance, particularly in older adults.

3. Clinical applicability and interpretation of diagnostic metrics

The study convincingly demonstrates that quantitative T1 mapping discriminates between CHS patients with and without MHE. However, as MHE status was determined prior to MRI, the reported AUC, sensitivity, specificity, PPV, and NPV describe discrimination within a predefined cohort rather than diagnostic accuracy in a real-world clinical population where MHE is suspected but not yet classified. The authors may wish to clarify this distinction in the Discussion to avoid overinterpretation. The current findings provide evidence of biomarker potential and pathophysiological relevance but do not yet establish clinical diagnostic performance.

4. Power analysis

The revised version now specifies that the sample size calculation was performed post hoc based on the study results. This clarification is appreciated, but the authors could consider noting this explicitly as a limitation in the Discussion. A post hoc calculation does not inform study design or recruitment targets and therefore provides a retrospective estimate of statistical power rather than evidence of adequate pre-study planning. A short acknowledgment of this would improve transparency.

Reviewer #2: All the questions from the reviewers well answered well and questions cleared and appreciate the changes in manuscript.

Reviewer #3: The revised manuscript titled “The role of quantitative T1 and T2 mapping for detecting minimal hepatic encephalopathy in chronic hepatic schistosomiasis patients” has been substantially improved and now demonstrates technical soundness and methodological rigor. The study design, data collection, and statistical analyses are clearly described and appropriately executed. The results are consistent with the stated objectives, and the conclusions are well supported by the data.

The authors have adequately addressed all previous reviewer comments, including clarification of the study design, sample size justification, MRI methodology, confounder adjustment, and inclusion of new tables and figures. The data availability requirements have been fully met.

Reviewer #4: Peer Reviewer Response Letter

Dear PLOS editors,

Thank you for the opportunity to review the revised version of the manuscript entitled “[Manuscript Title]” (Manuscript ID: [XXXX]). I have carefully examined the authors’ point to point responses and the revised manuscript.

Overall, I am quite pleased to note that the manuscript has been substantially improved. The authors have adequately addressed the major concerns raised in the previous review round in a thorough and satisfactory manner

The revised manuscript demonstrated improved conceptual clarity, stronger methodological transparency, expanded elaboration of discussion points and provided clarity in data presentation represented with well elaborated discussion points, including clarifications of methodology.

In my view the manuscript will be suitable for publication once the following minor comments are resolved by the authors:

Comment #1, “ The study design should be corrected with cross-sectional observational study, since the data is collected at a single point of time.”

Comment #2, “ Ensure that all the key words are stated in the abstract section including T2, NCT-A, DST, globus pallidus and China.”

Comment #3, “ The manuscript lacks background information and review of previous work on T2 mapping. It should be stated briefly in one or two lines.

Comment #4, “ Please include an operational definition of variables in your method section. Define,(Type B MHI,minimal hepatic encephalopathy, overt HE, and chronic CHS.)

Comment #5, “ Since tables and figures reflect your major findings, it will be better if you can present them in the result section rather than supplementary materials.

Comment #6, “ Please put a brief emphasis in your discussion that the findings for your research outcome is generated from.middle age and older CHS patients. ( N.B. The mean age of the patients in your study was approximately 70 and most of them were in their 5th, 6th and 7th decades of life.)

Comment #7, “ As the concluding statement of your discussion, you should highlight the quantitative role and diagnostic accuracy of T1 mapping in diagnosing MHE in middle aged and older CHS patients and its implications in the general population.

Comment #8, “ Make sure you state the key Messages of the scientific study in your conclusion section by raising the key points of MRI characterization; (a) T1 hyperintensity in the globus pallidus of CHS patients highlighting its diagnostic accuracy in differentiating MHE from non-MHI in CHS patients with mean age of 70, (b) emphasize its significance over conventional psychometric tests as a promising non-invasive and objective tool for detecting MHI in CHS patients and as potential surrogate marker for indicating the progression of chronic

**Do you want your identity to be public for this peer review?** For information about this choice, including consent withdrawal, please see our Privacy Policy

Reviewer #1: No

Reviewer #2: **Yes: ** Avinash Govinda Adiga

Reviewer #3: **Yes: ** Hala Awad Ahmed

Reviewer #4: **Yes: ** Robel Afeworki Habte

---

## [Author Response · Author response to Decision Letter 2]

19 Nov 2025

Thank you for your meticulous review and insightful feedback. We have carefully reviewed each comment and implemented comprehensive revisions accordingly. A detailed point-by-point response is provided below, with all modifications tracked in the text for your convenience.

Reviewer #1

Remaining minor points for clarification

1. Blinding procedures

The revised text specifies that two radiologists independently analyzed MRI data blinded to clinical information, which is commendable. However, it remains unclear whether they were also blinded to participants' MHE classification status. I suppose they were, but this could be explicitly stated if it was the case. Similarly, while the psychometric tests were administered using standardized instructions by trained examiners, it is not explicitly stated whether these examiners were aware of the patients' cognitive or clinical background. Clarifying these aspects, whether both MRI readers and psychometric assessors were fully blinded to group status, would help the reader better assess the internal validity of the findings and minimize concerns about potential measurement bias.

Response: Thank you for the comment. In our study, both MRI readers and psychometric test examiners were fully blinded to the participants’ MHE classification status and clinical information. We have clarified it in the text.

2. Education variable

Including education level in Table 1 is an important improvement, but the variable ("above/below primary school") remains rather coarse. Education is known to influence psychometric performance through factors such as literacy, occupational complexity, and cognitive stimulation, which may not be captured by this binary measure. The discussion could benefit from a short acknowledgment that this simplified categorization may not fully account for the effect of educational background on test performance, particularly in older adults.

Response: Thank you for the comment. We have added a statement in the Discussion section acknowledging this limitation.

3. Clinical applicability and interpretation of diagnostic metrics

The study convincingly demonstrates that quantitative T1 mapping discriminates between CHS patients with and without MHE. However, as MHE status was determined prior to MRI, the reported AUC, sensitivity, specificity, PPV, and NPV describe discrimination within a predefined cohort rather than diagnostic accuracy in a real-world clinical population where MHE is suspected but not yet classified. The authors may wish to clarify this distinction in the Discussion to avoid overinterpretation. The current findings provide evidence of biomarker potential and pathophysiological relevance but do not yet establish clinical diagnostic performance.

Response: Thank you for the comment. We have revised the Discussion to clarify the distinction between biomarker discrimination and clinical diagnostic validation. The new text highlights that further prospective validation in unclassified patient populations is needed to confirm real-world diagnostic utility.

4. Power analysis

The revised version now specifies that the sample size calculation was performed post hoc based on the study results. This clarification is appreciated, but the authors could consider noting this explicitly as a limitation in the Discussion. A post hoc calculation does not inform study design or recruitment targets and therefore provides a retrospective estimate of statistical power rather than evidence of adequate pre-study planning. A short acknowledgment of this would improve transparency.

Response: Thank you for the comment. We have revised the Discussion section to note this limitation, emphasizing that our post hoc analysis aimed to assess achieved power rather than guide sample size determination prospectively.

Reviewer #4

Comment #1, “ The study design should be corrected with cross-sectional observational study, since the data is collected at a single point of time.”

Response: Thank you for the comment. Accordingly, we have corrected the terminology throughout the manuscript to ensure consistency and accuracy.

Comment #2, “ Ensure that all the key words are stated in the abstract section including T2, NCT-A, DST, globus pallidus and China.”

Response: Thank you for the comment. We have revised the Abstract to ensure that all the key terms “T2, NCT-A, DST, globus pallidus, and China” are explicitly mentioned.

Comment #3, “ The manuscript lacks background information and review of previous work on T2 mapping. It should be stated briefly in one or two lines.

Response: Thank you for the comment. We have revised the Introduction section to include a short statement summarizing the role and prior findings of T2 mapping in hepatic encephalopathy research.

Comment #4, “ Please include an operational definition of variables in your method section. Define,(Type B MHI,minimal hepatic encephalopathy, overt HE, and chronic CHS.)

Response: Thank you for the comment. We have revised the Methods section to explicitly define Type B MHE, minimal hepatic encephalopathy (MHE), overt hepatic encephalopathy (HE), and chronic hepatic schistosomiasis (CHS), based on established diagnostic criteria and previous literature.

Comment #5, “ Since tables and figures reflect your major findings, it will be better if you can present them in the result section rather than supplementary materials.

Response: Thank you for the comment. We kept some of the tables in supplementary materials to maintain main text flow, with detailed data there for interested readers.

Comment #6, “ Please put a brief emphasis in your discussion that the findings for your research outcome is generated from.middle age and older CHS patients. ( N.B. The mean age of the patients in your study was approximately 70 and most of them were in their 5th, 6th and 7th decades of life.)

Response: Thank you for the comment. We have added a sentence in the Discussion highlighting that our findings may be most relevant to this demographic and that age-specific variations should be explored in future research.

Comment #7, “ As the concluding statement of your discussion, you should highlight the quantitative role and diagnostic accuracy of T1 mapping in diagnosing MHE in middle aged and older CHS patients and its implications in the general population.

Response: Thank you for the comment. We have revised the conclusion section to better highlight the quantitative diagnostic role of T1 mapping and clarify its implications for the general population.

Comment #8, “ Make sure you state the key Messages of the scientific study in your conclusion section by raising the key points of MRI characterization; (a) T1 hyperintensity in the globus pallidus of CHS patients highlighting its diagnostic accuracy in differentiating MHE from non-MHI in CHS patients with mean age of 70, (b) emphasize its significance over conventional psychometric tests as a promising non-invasive and objective tool for detecting MHI in CHS patients and as potential surrogate marker for indicating the progression of chronic

Response: Thank you for the comment. We have substantially revised the conclusion section to clearly articulate the main findings as suggested.

---

## [Decision Letter · Decision Letter 2]

5 Dec 2025

Dear Dr. Li,

Thank you for submitting your manuscript to PLOS ONE. After careful consideration, we feel that it has merit but does not fully meet PLOS ONE’s publication criteria as it currently stands. Therefore, we invite you to submit a revised version of the manuscript that addresses the points raised during the review process.

We look forward to receiving your revised manuscript.

Kind regards,

Ghada Abdrabo Abdellatif Elshaarawy, M.D

Academic Editor

PLOS One

Journal Requirements:

Additional Editor Comments:

It is advisable to add the figure of the ROC curve analysis and the table of multivariable logistic regression in the result section rather than the supplementary materials.

Reviewers' comments:

Reviewer's Responses to Questions

**Comments to the Author**

Reviewer #1: All comments have been addressed

Reviewer #3: All comments have been addressed

Reviewer #4: All comments have been addressed

2. Is the manuscript technically sound, and do the data support the conclusions?

Reviewer #1: Yes

Reviewer #3: Yes

Reviewer #4: Yes

3. Has the statistical analysis been performed appropriately and rigorously?

Reviewer #1: Yes

Reviewer #3: Yes

Reviewer #4: Yes

4. Have the authors made all data underlying the findings in their manuscript fully available?

Reviewer #1: Yes

Reviewer #3: Yes

Reviewer #4: Yes

5. Is the manuscript presented in an intelligible fashion and written in standard English?

Reviewer #1: Yes

Reviewer #3: Yes

Reviewer #4: Yes

Reviewer #1: The authors have fully addressed all previous comments with precision and care. I am satisfied with this revision and have no further comments.

Reviewer #3: The description of the blinding procedures has now been clearly articulated for both MRI readers and psychometric assessors, strengthening the internal validity of the study. The definitions of key clinical terms (Type B MHE, CHS, MHE, overt HE) are now appropriately included in the Methods section, improving precision and reproducibility. The manuscript also acknowledges important limitations, including the use of a simplified education variable and the post hoc sample size estimation.

The statistical analyses are appropriate and well executed. Normality testing, appropriate selection of statistical tests (t-test, Mann–Whitney U, chi-square), correlation analyses, and multivariable logistic regression were correctly applied. The ROC analysis is clear and supports the conclusions. The data provided in the public repository ensure transparency and compliance with PLOS data policies.

The revisions in the Discussion effectively distinguish between biomarker discrimination and real-world clinical diagnostic performance, avoiding over-interpretation. The authors have also added relevant context regarding T2 mapping and emphasized the study population’s age profile.

Overall, the manuscript is now clearly written, technically sound, and the conclusions are adequately supported by the data. The language is clear and intelligible, with only minor typographical issues that do not impede understanding.

Reviewer #4: (No Response)

**Do you want your identity to be public for this peer review?** For information about this choice, including consent withdrawal, please see our Privacy Policy

Reviewer #1: No

Reviewer #3: **Yes: ** Hala Awad Ahmed

Reviewer #4: **Yes: ** Robel Afeworki Habte

---

## [Author Response · Author response to Decision Letter 3]

5 Dec 2025

Thank you for your comments. We have carefully reviewed each comment and implemented comprehensive revisions accordingly. A detailed point-by-point response is provided below, with all modifications tracked in the text.

Additional Editor Comments:

It is advisable to add the figure of the ROC curve analysis and the table of multivariable logistic regression in the result section rather than the supplementary materials.

Response: Thank you for the comment. We have added a figure of the ROC curve analysis. And the table of multivariable logistic regression is now added in the result section.

---

## [Editor Report · Decision Letter 3]

9 Dec 2025

The role of quantitative T1 and T2 mapping for detecting minimal hepatic encephalopathy in chronic hepatic schistosomiasis patients

PONE-D-25-46086R3

Dear Dr. Li,

We’re pleased to inform you that your manuscript has been judged scientifically suitable for publication and will be formally accepted for publication once it meets all outstanding technical requirements.

Kind regards,

Ghada Abdrabo Abdellatif Elshaarawy, M.D

Academic Editor

PLOS One
---

## [Editor Report · Acceptance letter]

PONE-D-25-46086R3

PLOS One

Dear Dr. Li,

I'm pleased to inform you that your manuscript has been deemed suitable for publication in PLOS One. Congratulations! Your manuscript is now being handed over to our production team.

Kind regards,

on behalf of

Dr. Ghada Abdrabo Abdellatif Elshaarawy

Academic Editor

PLOS One